# Use of Metabolomic Profiling to Understand Variability in Adiposity Changes Following an Intentional Weight Loss Intervention in Older Adults

**DOI:** 10.3390/nu12103188

**Published:** 2020-10-19

**Authors:** Ellen E. Quillen, Daniel P. Beavers, Anderson O’Brien Cox, Cristina M. Furdui, Jingyun Lee, Ryan M. Miller, Hanzhi Wu, Kristen M. Beavers

**Affiliations:** 1Internal Medicine, Section on Molecular Medicine, Wake Forest School of Medicine, Winston-Salem, NC 27157, USA; equillen@wakehealth.edu (E.E.Q.); cfurdui@wakehealth.edu (C.M.F.); jilee@wakehealth.edu (J.L.); hwu@wakehealth.edu (H.W.); 2Biostatistics and Data Science, Wake Forest School of Medicine, Winston-Salem, NC 27101, USA; dbeavers@wakehealth.edu; 3Proteomics and Metabolomics Shared Resource, Comprehensive Cancer Center, Wake Forest School of Medicine, Winston-Salem, NC 27157, USA; aocox@wakehealth.edu; 4Internal Medicine, Section on Gerontology and Geriatric Medicine, Wake Forest School of Medicine, Winston-Salem, NC 27157, USA; millerr@wfu.edu; 5Health and Exercise Science, Wake Forest University, Winston-Salem, NC 27109, USA

**Keywords:** metabolomics, weight loss, body composition, aging, heterogeneity

## Abstract

Inter-individual response to dietary interventions remains a major challenge to successful weight loss among older adults. This study applied metabolomics technology to identify small molecule signatures associated with a loss of fat mass and overall weight in a cohort of older adults on a nutritionally complete, high-protein diet. A total of 102 unique metabolites were measured using liquid chromatography-mass spectrometry (LC-MS) for 38 adults aged 65–80 years randomized to dietary intervention and 36 controls. Metabolite values were analyzed in both baseline plasma samples and samples collected following the six-month dietary intervention to consider both metabolites that could predict the response to diet and those that changed in response to diet or weight loss.Eight metabolites changed over the intervention at a nominally significant level: D-pantothenic acid, L-methionine, nicotinate, aniline, melatonin, deoxycarnitine, 6-deoxy-L-galactose, and 10-hydroxydecanoate. Within the intervention group, there was broad variation in the achieved weight-loss and dual-energy x-ray absorptiometry (DXA)-defined changes in total fat and visceral adipose tissue (VAT) mass. Change in the VAT mass was significantly associated with the baseline abundance of α-aminoadipate (*p* = 0.0007) and an additional mass spectrometry peak that may represent D-fructose, myo-inositol, mannose, α-D-glucose, allose, D-galactose, D-tagatose, or L-sorbose (*p* = 0.0001). This hypothesis-generating study reflects the potential of metabolomic biomarkers for the development of personalized dietary interventions.

## 1. Introduction

The prevalence of obesity and its detrimental health effects are increasing rapidly among older adults [1,2]. Medical complications associated with excess fat mass highlight the need to treat obesity in this age group; [3] however, recommendation for intentional weight loss remains controversial [4,5]. Reluctance stems, at least in part, from the loss of lean mass known to accompany overall weight loss (10–50% of total tissue [6,7]), and the potential exacerbation of age-related disability risk. Encouragingly, the data from randomized controlled trials (RCTs) support immediate muscle strength and function gains among older adults following lifestyle-based weight loss—particularly when structured exercise is included as an intervention component—despite lean mass loss [8,9]. 

Accordingly, current geriatric obesity treatment guidelines encourage weight loss therapies that minimize lean, while maximizing fat, mass loss for older adults with obesity [10]. Change in body composition with caloric restriction-induced weight loss appears modifiable through diet, with the amount of dietary protein consumed during caloric restriction identified as a key determinant in lean mass preservation [11]. Indeed, meta-analytic data show that older adults who consume higher levels of protein during weight loss retain more lean mass in comparison to normal protein diets (losses of 21–22% vs. ≥30%) [12]. In agreement, the data from our group show older adults following a hypocaloric, nutritionally complete, higher protein meal plan experience smaller average lean (~13%) vs. fat (~87%) mass losses; however, the amount and location of fat mass loss was noted to be more variable [13,14]. A better understanding of the variability in adiposity changes as a means to optimize fat mass loss in this context confers high clinical utility.

Metabolites (i.e., small molecules including sugars, amino acids, and vitamins which are both reactants and products of metabolic processes in the body) are particularly attractive biomarkers for understanding variability in response to dietary interventions. Metabolites can be collected from blood, urine, or other biofluids and are sensitive reflections of both intrinsic and extrinsic changes in nutrition and metabolism [15,16]. Prior studies have shown that baseline metabolomic profiles are associated with body composition responsiveness [17,18] and that select metabolites, including branch chain amino acids, may change following a weight loss intervention [15]. Herein, we sought to identify the plasma metabolites associated with the changes in total and visceral fat mass among older adults following a hypocaloric, nutritionally complete, higher protein meal plan. Although primarily meant to be hypothesis generating, these results may serve as preliminary data to design targeted dietary interventions for fat loss among older adults. 

## 2. Materials and Methods

### 2.1. Study and Participant Descriptions

This analysis utilizes data from The Medifast^®^ for Seniors Study (NCT02730988), a six month RCT conducted at Wake Forest University (recruiting from 18 September 2015 to 14 September 2016), designed to compare the effects of weight loss (WL), achieved by following a hypocaloric, nutritionally complete, higher protein meal plan, vs. weight stability (WS) on mobility and body composition in 96 older adults (54–79 years) with obesity (30–40 kg/m^2^). This study was approved by the Wake Forest School of Medicine Institutional Review Board (IRB: 33,428 and 54,086), and all participants provided written informed consent prior to enrolment. Primary outcome papers, including study design and dietary details, were previously published [13,14,19,20]. For the present analyses, we evaluated the metabolomic data from plasma samples drawn at baseline and at the end of the six month dietary intervention for 74 individuals with complete Dual-Energy X-Ray Absorptiometry (DXA)-derived body composition data.

### 2.2. Metabolite Extraction

Metabolites were extracted from 50 µL of plasma spiked with 10 µL of internal standard, 2-(N-morpholino) ethanesulfonic acid (MES) solution using a standard extraction method of four volumes of cold methanol and incubation on ice for 30 min. After centrifugation at 18,000× *g* for five minutes, the supernatant was dried under vacuum and reconstituted in ultrapure water for liquid chromatography–mass spectrometry (LC–MS) analysis.

### 2.3. Broad Metabolomics Data Generation and Processing

The LC–MS consisted of a Q Exactive HF hybrid quadrupole–orbitrap mass spectrometer (Thermo Scientific) and a Vanquish ultra-high pressure liquid chromatography (UHPLC) system (Thermo Scientific). To identify a broader range of metabolites, the samples were analyzed on two different columns, a Hypersil GOLD pentafluorophenyl (PFP) column and an Accucore Vanquish C18+ column. A linear gradient was employed for chromatographic separation using 100% water (mobile phase A) and 90% acetonitrile (mobile phase B) both of which contained 0.1% formic acid and 10 mM ammonium formate. Data were acquired by collecting full mass spectra (MS1) using polarity switching (positive/negative) at a resolution of 150 K. 

Molecular ion peaks were extracted and integrated within the Mass Spectrometry Metabolite Library of Standards (MSMLS) Discovery software (IROA Technologies) in combination with customized compound libraries prepared using a Mass Spectrometry Metabolite Library (Sigma-Aldrich). To eliminate redundancy in compound identification, the most abundant ion in each peak was selected and the peak area was then normalized to the total ion current (TIC) for relative quantification. We quantile normalized and log transformed the data to reduce the bias caused by a small number of highly abundant metabolites [21]. Because the missing data from a mass spectrometer are not generally representative of a true zero where values are present for other samples, we replaced the missing values with half of the minimum detected value in the full data set. The analysis was performed in parallel using probabilistic quotient normalization (PQN), however, this method was unable to remove residual batch effects from the run order of the samples. As PQN and quantile normalization produce similar accuracy in untargeted LC/MS data of this sample size [22], all results presented reflect findings with quantile-normalization.

### 2.4. Body Weight, Composition, and Fat Distribution

Body weight and composition were measured at baseline and six months. Baseline weight was measured to the 10th decimal without shoes and outer garments using a calibrated scale (Detecto 758C Weight Indicator; Webb City, MO). Height was obtained without shoes to within 0.25 inches using a QuickMedical 235D Heightronic Digital Stadiometer (Issaquah, WA). Body mass index (BMI) was calculated as weight (kg)/height (m^2^). Dual-energy x-ray absorptiometry (DXA) scans were used to determine the total body, lean, and fat masses, with visceral adipose tissue (VAT) derived using the CoreScan algorithm (GE Medical Systems, Madison, WI, USA) [23]. All scans were performed on the same machine and by the same technician, following the manufacturer recommendations for patient preparation and positioning. Coefficients of variation from repeated measurements at our institution are <1.0% for total body, lean, and fat masses.

### 2.5. Statistical Analyses

For participants with complete DXA data (*n* = 74), baseline metabolomic data were evaluated using principal components analysis to identify any stratification associated with age, gender, race, or technical artifacts (i.e., LC–MS run order). Demographic characteristics, including age, gender, and race were assessed by the participant report at baseline.

For 102 unique MS peaks, mean differences in the WL and WS groups were evaluated using Welch’s *t*-tests to account for differences in variance. Within the WL group (*n* = 38), we evaluated the differences in the loss of total weight, total fat mass, and VAT mass associated with either metabolite abundance at baseline or change in metabolite abundance over the six month intervention using linear regression. 

We also considered the percent change in weight and body composition to account for baseline differences. Change in the weight, total fat mass, and VAT mass all showed substantial but non-significant differences between men and women at baseline, so we considered both models containing all participants and gender-stratified models. Males were 23.5 kg heavier on average with 3.6 kg more total fat mass and 1.9 kg more VAT mass. The relationship between demographic variables and individual metabolites or principal components of metabolomic variation were tested and found to be non-significant except with regard to gender. Based on this, age and race were not incorporated into subsequent models. We undertook these analyses solely in the WL group because the diet provided to this group was expected to cause broad shifts in the nutrient profile irrespective of weight loss that could confound associations if the WS group was included. All analyses were performed in R [24]. Throughout, MS peaks were considered independent of one another so multiple testing was addressed through the use of the Bonferroni-corrected critical *p*-value of 0.0007.

We implemented a pathway over-representation analysis in Impala [25] to contextualize the top 25 most associated metabolites. Pathways were considered over-represented if they contained at least 4 of the top 25 most strongly associated metabolites and had a false discovery rate *q* < 0.01. To address cases where a peak could not be unambiguously linked to a single metabolite, ten groups of metabolites were tested with a single metabolite randomly selected to represent each peak. This did not alter the significant pathways.

## 3. Results

### 3.1. Intervention-Related Changes in Body Composition and Sample Characteristics

Body composition changes in the Medifast^®^ for Seniors Study have been published previously [13,14]. Seventy-four study participants had complete pre- and post-intervention DXA measures and plasma samples available for the metabolomics analysis. Demographic and body composition data for the subset of individuals with complete metabolomics and body composition data are presented in Table 1. Individuals were 65–80 years of age, 58% female, and 76% white with a starting BMI between 30 and 42 kg/m^2^. Over the course of the six month period, the total body mass was significantly reduced in the WL group as compared with the WS group, with 87% of total mass lost as fat. Significant reductions in VAT were also observed in the WL but not in the WS group. In contrast, lean mass loss was much more modest, and no differential treatment effect was observed between groups. Figure 1 highlights the broad distribution in weight, total fat, and VAT mass lost over the intervention compared to change in lean mass. 

### 3.2. Metabolomics Data Generation and Normalization

We detected 102 unique mass spectrometry feature (MS1) peaks present above background levels in at least 10 samples corresponding to 118 metabolites due to some overlap in the chromatographic separation. After normalization, three individuals were removed from the sample for non-standard distributions of metabolites identified in the principal components analysis (PCA). This appears to have been driven by the number of missing metabolites in these samples.

### 3.3. Metabolites Associated with Participant Baseline Characteristics

In the baseline plasma samples, we identified only two metabolites (creatine and 5,6-dihydrouracil) differing by gender, and none that differed by age or race. We noted that all participants included in the metabolomics analysis were between 65 and 80 years old, so the effect of age on metabolite variability would be muted. 

### 3.4. Metabolomic Differences in Weight Loss vs. Weight Stable

After six months on the intervention diet, five metabolites—D-pantothenic acid, L-methionine, nicotinate, aniline, and melatonin—decreased in the WL group but not in the WS group, while three metabolites increased—deoxycarnitine, 6-deoxy-L-galactose, and 10-hydroxydecanoate using a nominal α = 0.05 (Figure 2). The heatmap in Figure 3 shows the clustering of WL and WS plasma samples and metabolites at six months based on differentially expressed metabolites. We note that the clustering of the WL and WS groups are not mutually exclusive across the top axis. 

### 3.5. Metabolites Associated with Change in Weight and Fat Mass 

We evaluated change in metabolite abundance for all 102 MS1 peaks from baseline to six months vs. percent change in weight, total fat mass, or VAT, to evaluate metabolites that may be produced during weight loss. At a Bonferroni correction *p* < 0.0007, no metabolites were significantly associated.

When comparing percent change in weight and fat mass with baseline variables to identify potential predictive biomarkers, one MS peak was marginally significant in the full analysis, α-aminoadipate (R^2^ = 0.29, *p* = 0.0007). This peak was also nominally significant in the female-specific analysis and showed a consistent direction of effect but no significant association in the male-specific analysis. In the female-specific analysis, metabolite abundance at one MS peak is significantly associated at *p* = 0.0001 (R^2^ = 0.46). This peak is linked to eight potential metabolites present in the Mass Spectrometry Metabolite Library (D-fructose, myo-inositol, mannose, α-D-glucose, allose, D-galactose, D-tagatose, or L-sorbose), which cannot be distinguished by mass or chromatography profiles. Figure 4 shows the relationship between each peak intensity in WL participants at baseline vs. VAT change over intervention.

### 3.6. Pathways Associated with VAT Change

Pathways from several databases indexed in the Impala software are enriched for metabolites associated with VAT change in our sample (Table 2). Among these is a large general metabolism pathway containing 21 of the top 25 VAT-associated metabolites. However, others are more specific including several pathways involved in protein and amino acid metabolism. Seven of the pathways are represented by four amino acids—tyrosine, proline, methionine, and glutamic acid—which were in lower abundance at baseline among individuals who lost a greater VAT mass. These amino acids are common in commercially available whey and soy proteins which are a major component of the Medifast^®^ diet [26].

## 4. Discussion

The goal of this study was to identify the metabolites associated with inter-individual differences in total and regional fat loss during a weight loss intervention in older adults known to produce a lean mass sparing effect. When compared to the weight stable group, three metabolites increased across the intervention (deoxycarnitine, 6-deoxy-l-galactose, and 10-hydroxydecanoate) while five metabolites decreased (D-pantothenic acid, L-methionine, nicotinate, aniline, and melatonin). Additionally, we report an association between greater baseline levels of α-aminoadipate (also called 2-aminoadipic acid) and myo-inositol or a carbohydrate metabolite (α-D-glucose, allose, mannose, D-galactose, D-fructose, and L-sorbose) with larger decreases in visceral adipose tissue. This information is a critical first step in developing personalized approaches to optimize weight loss interventions for older adults.

Given the dearth of literature concerning metabolites that change in response to change in diet, it is difficult to hypothesize about the origins of diet-associated metabolite increases that were observed in our study, but it is possible that they are related to the long-term change in diet (e.g., caloric restriction or dietary composition). Interestingly, deoxycarnitine has previously been shown to increase following weight loss interventions in animal models [27]. Since deoxycarnitine is the final intermediate for endogenous L-carnitine biosynthesis, it is possible that L-carnitine biosynthesis was elevated in the weight loss group, resulting in greater decreases in visceral adipose tissue [28]. 

The amino acid pathways identified in our study align with previous observations and may reflect improvements in metabolic perturbations following dietary intervention [29,30,31]. However, despite observing decreases in amino acids and visceral adipose tissue during weight loss, previous research examining the relationship between visceral fat depots and amino acids remains inconclusive [32,33,34]. While individual amino acid levels were not linked to changes in VAT in our study, several pathways related to protein and amino acid metabolism are enriched for nominally significant metabolites. This over-representation is largely driven by four amino acids at high abundance in the soy and whey proteins. The non-significant association of lower amino acid levels at baseline with a greater VAT loss in our sample may reflect a greater benefit of high-protein dietary weight loss interventions for older adults with low protein consumption at baseline.

The weight loss group also displayed declines in vitamin B family metabolites (pantothenic acid and nicotinate), which may have been a function of decreased food intake as part of enrolling in the intervention. Additional vitamin B6 responsive metabolites were also over-represented in our VAT pathway analysis. Interestingly, decreases in melatonin occurred within the weight loss group, which has displayed promising effects for improving body composition and may ameliorate the consequences of obesity [35,36,37]. 

To date, few studies have examined if baseline metabolomic signatures are associated with weight loss outcomes [17,18]. Stroeve et al. [17] suggested that 57% of weight loss success could be identified from baseline metabolomic signatures in morbidly obese men and women, which included acetoacetate, triacylglycerols, phosphatidylcholines, amino acids, creatine and creatinine. Additional observations suggest that baseline xylitol and uridine levels were inversely associated with weight loss, whereas 2-aminobutyric acid and glyceric acid were positively associated with weight loss ≥10% over a 1 year intervention [18]. 

The metabolites identified in our study mirror and extend recent observations between these metabolites (e.g., myo-inositol and glycolytic/gluconeogenic intermediates) and visceral adipose tissue [38]. When previous research stratified weight loss participants into high- or low-responders post-intervention (≥7.2 kg and ≤5.2 kg weight loss, respectively); myo-inositol, among other metabolite levels, were greater in the high-responders at baseline [39]. Additionally, previous research examining myo-inositol supplementation has also observed an attenuation of metabolic abnormalities [40,41,42]. Regarding the elevated carbohydrate metabolites, obesity has been shown to be associated with increased levels of these compounds, and the metabolomic signature of carbohydrate metabolism for obesity and diabetes are quite similar [43]. This is further bolstered by our association of elevated α-aminoadipate levels at baseline with greater VAT mass loss. This metabolite is an oxidation product considered a biomarker of early stage diabetes and for cardiometabolic complications among individuals with diabetes [44,45].

Personalized interventions to maximize overall improvements in health is the ultimate goal of weight-loss research in older adults. Because of its association with an increased risk of cardiometabolic diseases [46] (independent of BMI), the identification of biomarkers associated with a loss of VAT may be particularly important in targeting dietary approaches that can minimize lean mass loss while maximizing cardiometabolic benefit. Given the previous links between visceral adipose tissue and altered metabolic effects [38,47], the elevated carbohydrate metabolites at baseline may identify individuals with greater metabolic irregularity (e.g., those with decreased metabolic health). While no study participants had diabetes, these baseline metabolites may represent individuals with the most to gain cardiometabolically from a weight loss intervention.

While studies of weight loss-associated metabolomic biomarkers have increased rapidly in recent years, we are among a limited number of studies to report baseline metabolite profiles associated with response to dietary interventions [15,17,18]. Our findings are limited by the fact that the analyses were performed post hoc and that the study was not originally designed or powered to evaluate metabolite associations. However, the identification of baseline glucose metabolites linked to variation in VAT loss provides an avenue to identify individuals who may benefit the most from a lower-sugar dietary prescription. Future studies incorporating lipidomics or untargeted metabolomics could further elucidate the differences at baseline that predict response to interventions. 

## Figures and Tables

**Figure 1 nutrients-12-03188-f001:**
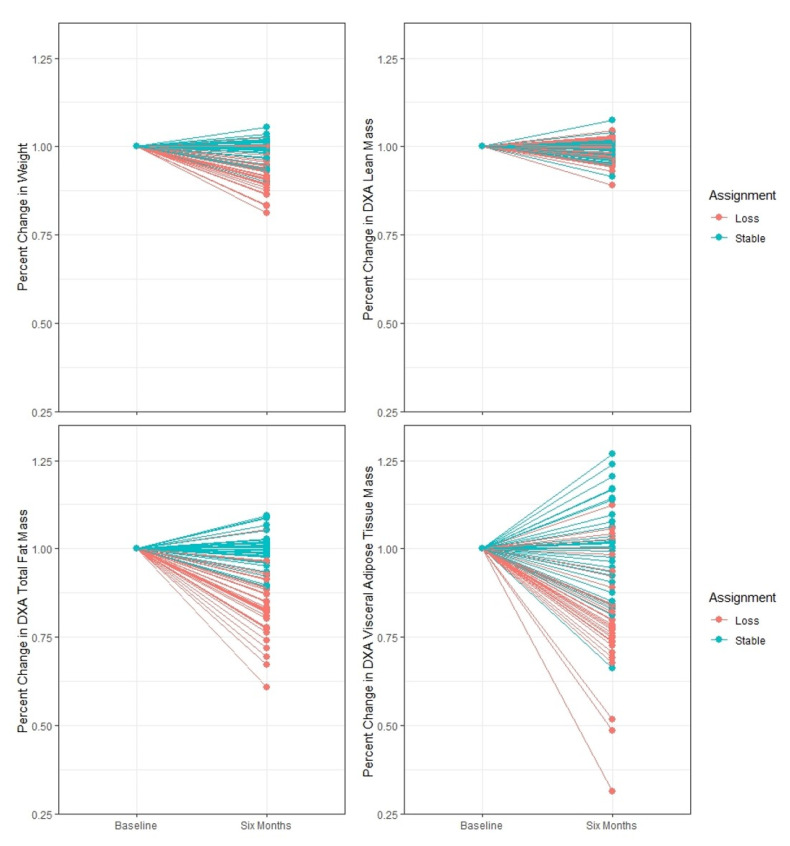
Change in the weight and body composition. From baseline to six months, the percent change in total body weight, lean mass, fat mass, and visceral adipose tissue (VAT) mass in individuals randomized to the weight loss (orange; *n* = 38) and weight stable (teal; *n* = 36) groups.

**Figure 2 nutrients-12-03188-f002:**
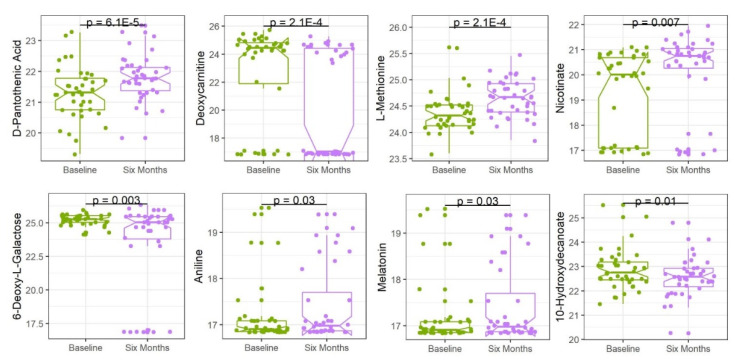
Change in mean metabolite abundance in weight loss group. The distribution with mean and inter-quartile ranges are shown for eight metabolites with nominally significant *p*-values in Welch’s *t*-tests between baseline and the end of the six-month weight loss intervention.

**Figure 3 nutrients-12-03188-f003:**
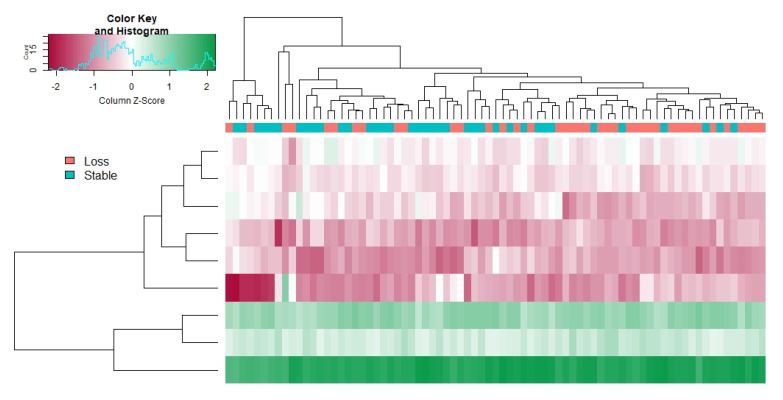
Heatmap of differentially expressed metabolites between weight loss (WL) and weight stable (WS) groups at the end of the intervention. Based on metabolites with nominally significant (*p* < 0.05) Welch’s *t*-test results comparing the groups at the six month time point, this heatmap shows clustering of highly expressed (green) and lowly expressed (red) metabolites. The dendrogram at the top clusters individuals based on the similarity of their metabolite profiles with individuals in the WL group in orange and the WS group in teal.

**Figure 4 nutrients-12-03188-f004:**
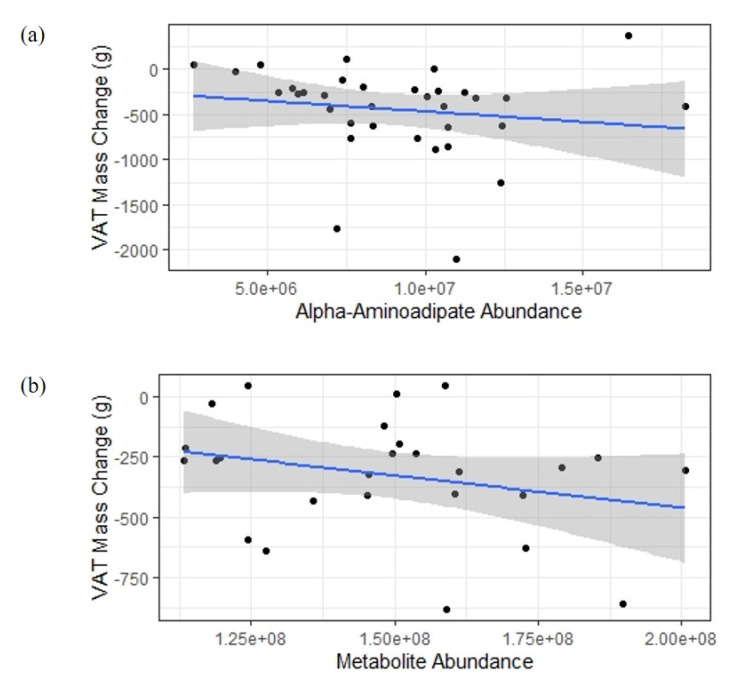
Scatter plot of baseline metabolite abundance vs. change in visceral adipose tissue (VAT). (**a**) Abundance of α-aminoadipate at baseline in all individuals randomized to weight loss (WL) vs. VAT mass change in grams following the intervention. (**b**) Abundance of the indeterminate metabolite at baseline vs. VAT mass change for females in the WL group only.

**Table 1 nutrients-12-03188-t001:** Demographic characteristics and baseline and six-month change in body composition for weight loss and weight stable groups in metabolomics sample. BMI = body mass index, VAT = visceral adipose tissue.

	Randomization Grouping		
	Weight Loss	Weight Stable	Student’s T *p*
**Group N (%)**			
**Female**	28 (74%)	27 (75%)	-
**Male**	10 (26%)	9 (25%)	-
**White**	27 (71%)	29 (81%)	-
**Black**	11 (29%)	7 (19%)	-
**Mean (Standard Deviation)**			
**Age at Baseline**	71.4 (4.0)	68.9 (3.1)	-
**BMI at Baseline**	34.8 (3.6)	35.5 (3.2)	-
**Change in Weight, kg**	−7.9 (5.5)	−0.5 (2.9)	6 × 10^−10^
**Females**	−6.9 (3.7)	−0.05 (2.7)	<2 × 10^−16^
**Males**	−10.7 (7.7)	−2.0 (2.9)	6 × 10^−5^
**Change in Lean Muscle Mass (kg)**	−2.9 (3.4)	−0.5 (1.0)	0.12
**Females**	−0.5 (1.1)	−0.4 (1.3)	0.59
**Males**	−2.2 (3.0)	−0.2 (1.7)	0.01
**Change in Total Fat Mass (kg)**	−6.9 (4.2)	−0.02 (2.0)	2 × 10^−13^
**Females**	−7.9 (3.3)	−5.8 (1.9)	<2 × 10^−16^
**Males**	−8.0 (5.7)	−1.2 (2.1)	4 × 10^−5^
**Change in VAT Mass (kg)**	−0.8 (0.9)	−0.7 (0.3)	1 × 10^−4^
**Females**	−0.4 (0.3)	0.08 (0.2)	2 × 10^−14^
**Males**	−0.6 (1.0)	−0.2 (0.4)	0.12

**Table 2 nutrients-12-03188-t002:** Pathways significantly enriched for metabolites associated with change in visceral adipose tissue (VAT).

Pathway Name	Total Metabolites	False Discovery Rate (FDR) q	Database	VAT-Associated Metabolites
Metabolism	1384	0.009	Reactome	L-Proline, Urate, Alpha-Aminoadipate, N-Acetyl-L-Aspartic Acid, N-Acetyl-D-Galactosamine, L-Glutamic Acid, Myo-Inositol, 4-Hydroxy-L-Proline, L-Tyrosine, 4-Hydroxy-2-Quinolinecarboxylic Acid, D-Glucuronic Acid, Urocanate, 5,6-Dihydrouracil, Uridine, Creatine, Alpha-D-Glucose, Paraxanthine, Mannose, N-Acetyl-D-Glucosamine, D-Galactose, D-Fructose
SLC-mediated transmembrane transport	166	0.028	Reactome	L-Proline, Urate, L-Glutamic Acid, 4-Hydroxy-L-Proline, L-Tyrosine, Myo-Inositol, Uridine, Alpha-D-Glucose, Mannose, D-Fructose
Metabolism of amino acids and derivatives	285	0.003	Reactome	L-Proline, Alpha-Aminoadipate, N-Acetyl-L-Aspartic Acid, L-Glutamic Acid, 4-Hydroxy-L-Proline, L-Tyrosine, Creatine, Urocanate, 4-Hydroxy-2-Quinolinecarboxylic Acid
Histidine, lysine, phenylalanine, tyrosine, proline, and tryptophan catabolism	91	0.002	Reactome	L-Proline, Alpha-Aminoadipate, L-Glutamic Acid, 4-Hydroxy-L-Proline, L-Tyrosine, Urocanate, 4-Hydroxy-2-Quinolinecarboxylic Acid
Aminoacyl-tRNA biosynthesis	52	0.034	KEGG	L-Tyrosine, L-Proline, L-Methionine, L-Glutamic Acid
Protein digestion and absorption	47	0.028	KEGG	L-Tyrosine, L-Proline, L-Methionine, L-Glutamic Acid
Central carbon metabolism in cancer	37	0.015	KEGG	L-Tyrosine, L-Proline, L-Methionine, L-Glutamic Acid
S-methyl-5-thio-α-D-ribose 1-phosphate degradation	35	0.014	HumanCyc	L-Tyrosine, L-Proline, L-Methionine, L-Glutamic Acid
Leukotriene biosynthesis	30	0.012	HumanCyc	L-Tyrosine, L-Proline, L-Methionine, L-Glutamic Acid
γ-glutamyl cycle	29	0.012	HumanCyc	L-Tyrosine, L-Proline, L-Methionine, L-Glutamic Acid
tRNA charging	24	0.009	HumanCyc	L-Tyrosine, L-Proline, L-Methionine, L-Glutamic Acid
Vitamin B6-dependent and responsive disorders	18	0.036	Wikipathways	L-Proline, L-Glutamic Acid, Alpha-Aminoadipate

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
