# Peer review of "Use of Metabolomic Profiling to Understand Variability in Adiposity Changes Following an Intentional Weight Loss Intervention in Older Adults"

_nutrients, 2020, doi:10.3390/nu12103188_

Round 1
Reviewer 1 Report
This paper presents new interesting data on the identification of metabolites associated with loss of total fat mass and visceral fat mass in a sample of older adults. The following points need to be addressed:
Materials and Methods
Body weight, Composition, and Fat-Distribution
- The measurement of weight to the 10th decimal is meaningless. Modify this indication and give more information on anthropometric methods and references (see, for example, the description of anthropometric methodologies in Zaccagni et al. Int. J. Environ. Res. Public Health 2020, 17, 4273; doi:10.3390/ijerph17124273). The BMI has not been mentioned
Statistical analises
2.indicate the pre-chosen significance level (p).
Results
- lines 145-149: explain the two values in brackets ( for example, those close to the decrease of the total body mass). The same for the following values.
- as is well known, males and females differ significantly in body composition. Despite the small size of the samples considered, it would be more appropriate to provide the results separately by sex.
- lines 166-167: you reported that “all participants are between 65 and 80 years old”, while in the sample description you indicated a different age range (54-79). Clarify this.
Reviewer 2 Report
The paper investigates metabolic differences that are associated to weight loss. It is very well written.
However, I feel that the analyses are uninspired. It is a very standard metabolomics paper which is lacking pathway analyses. It would be good to include them so that the discussion is not just focused on single metabolites.
The quantile normalization of the metabolites is not one I would ever choose. Especially for untargeted data, which is presented in the paper, the probabilistic quotient normalization (PQN) followed by log2 transformation is recommended.
Round 2
Reviewer 2 Report
Thank you for including the PQN and the pathway analyses.